# Aphid Behavior on *Amaranthus hybridus* L. (Amaranthaceae) Associated with *Ocimum* spp. (Lamiaceae) as Repellent Plants

**Boni Barthélémy Yarou [1,2,*]**, **Aimé H. Bokonon-Ganta [3]**, **François J. Verheggen [2]**, **Georges C. Lognay [2,*]** and **Frédéric Francis [2]**

1   Institut National des Recherches Agricoles du Bénin, Sous-Programme Cultures Maraîchères, Recette Principale, Cotonou, Godomey, Route IITA, Cotonou 01 BP: 884, Benin

2   Functional and Evolutionary Entomology, TERRA Research Center, Gembloux Agro-bio Tech, University of Liege (ULg) Passage des Déportés, 2 BE-5030 Gembloux, Belgium; fverheggen@uliege.be (F.J.V.); frederic.francis@uliege.be (F.F.)

3   Laboratoire d'Entomologie Agricole, Faculté des Sciences Agronomiques, Université d'Abomey-Calavi, Cotonou 03 BP: 2819, Benin; aimehbg@gmail.com

*   Correspondence: boniyarou1981@gmail.com (B.B.Y.); georges.lognay@uliege.be (G.C.L.)

**Abstract:** Various plant species contain biocidal and/or semiochemical components. These can be used for managing insect pests, in order to reduce the use of synthetic pesticides and to improve the quality of vegetable crops. This study was conducted to assess the effect of repellent plants *Ocimum gratissimum* L. and *Ocimum basilicum* L. on aphids *Aphis craccivora* Koch, *Aphis fabae* Scopoli and *Myzus persicae* Sulzer when they are associated with *Amaranthus hybridus* L. plants. The results have shown that in the two approaches tested—*Ocimum* sp. plants surrounded by *A. hybridus* plants and the dual-choice test—the number of aphids on the *A. hybridus* plant associated with either *O. gratissimum* or *O. basilicum* was significantly less significant compared to the *A. hybridus* alone. This first study on the association between *A. hybridus* and *Ocimum* spp. shows that the *Ocimum* species might be used as an alternative method for controlling aphids in order to avoid the use of synthetic pesticides on *Amaranthus*. The ability of *Ocimum* spp. to repel pests can make it an important companion plant for farmers, because those plants can not only be used to control pests, but they can also be harvested, providing a direct economic return.

**Keywords:** aphids; basil; pest control; amaranth; repellent activity; companion plant

## 1. Introduction

Aphids are pests that cause economically significant crop losses on many vegetable crops. To control these aphid populations, synthetic pesticides are increasingly used, especially in developing countries [1]. However, there is, nowadays, a growing public concern about the harmful effects of pesticides on humans and non-target organisms, in addition to the resistance of many pests to some commonly used insecticides [2,3]. Together, these problems have motivated the development of environmentally friendly alternative means to manage pests. Biological control using natural enemies is one of these alternatives to control crop pests. However, although this form of control strategy can potentially be successful at a low cost [4], it remains difficult to implement by small-scale farmers in developing countries. In this context, cheap, plant-based solutions can play an important role. Using plants with pesticidal properties to protect crops against pests can have advantageous effects for humans, such as reduced environmental degradation and increased food safety [5]. Indeed, botanical extracts have been reported to be effective on pests and pathogens [6,7]. These plants can be used to

protect crops by keeping pests away and then avoiding potential damage [8,9]. The biocidal activity of these plants is generally attributed to volatile organic compounds (VOC) that are emitted and can directly affect herbivorous pests due to their toxic, repellent or dissuasive properties [10,11].

*Ocimum* spp (Lamiaceae), or basils, are important plants with pesticidal effects, whose biocidal effect has been reported mainly on stored-product pests and malaria vectors [12–14]. Some studies indicated that their qualities as repellent or companion plants decreased pest abundance on crops. In a semi-field experiment, *O. americanum* L. essential oil had a repellent effect on *Agrotis ipsilon* Hufnagel (Lepidoptera: Noctuidae) [15]. Testing *O. basilicum* as a banker plant in greenhouse tomato crops led to fewer recorded pests [16]. Similarly, near to the tomato plants, *Ocimum* spp plants (*O. gratissimum* and *O. basilicum*) or their essential oils (through diffusers) reduced the oviposition of *Tuta absoluta* Meyrick (Lepidoptera: Gelechiidae) [17]. Intercropping of *O. basilicum* with *Gossypium barbadense* L. (Malvaceae) was also found to reduce pest abundance [18]. Introducing *O. basilicum* in *Vicia faba* L. (Fabaceae) reduced infestation levels of the bean aphid–*Aphis fabae* Scopoli (Hemiptera: Aphididae) [19]. A repellent effect was also reported on cabbage pests *Phyllotreta sinuata* Steph. (Coleoptera: Chrysomelidae), *Hellula undalis* Fabricius (Lepidoptera: Crambidea), *Spodoptera litura* Fabricius (Lepidoptera: Noctuidae), *Spodoptera littoralis* Fabricus (Lepidoptera: Noctuidae) and *Plutella xylostella* (Lepidoptera: Plutellidae) when this crop was intercropped with *Ocimum* species [9,20]. *Ocimum basilicum* reduced pest abundance by 23% on *Abelmoschus esculentus* L. (Malvaceae) compared to the control, and also reduced the use of synthetic pesticides. In an orchard ecosystem, it was reported that planting *Ocimum* spp between trees can reduce pest levels and also attract natural enemies, including Coccinellidae, Syrphidae, Chrysopidae and Phytoseiidae [21–23].

The aim of the present study was to evaluate the repellent effect of *O. basilicum* and *O. gratissimum* on the aphids (Hemiptera: Aphididae) *Aphis craccivora* Koch (Cowpea aphid), *A. fabae* Scop. and *Myzus persicae* Sulzer (Green peach aphid). The experiments were carried out on *Amaranthus hybridus* L. (Amaranthaceae), one of the most popular vegetables crops in tropical Africa [24] and highly attacked by aphids [25–27]. The *Ocimum* species are also used as vegetables in several West African countries [28]. The outcome of these experiments shows that *Ocimum* could be used as a companion plant for pest management.

## 2. Materials and Methods

### 2.1. Plants and Aphids Rearing

Seeds of *Amaranthus hybridus* (Ah), *O. gratissimum* (Og) and *O. basilicum* (Ob) were provided by the Vegetable Crops Research Program of the National Institute of Agricultural Research of Benin (INRAB), West Africa. The plants were grown under greenhouse condition ($25 \pm 5$ °C, 50%–70% relative humidity, 16:8-h light: dark) in plastic pots ($8 \times 8 \times 9$ cm) filled with potting soil (VP113BIO, Peltracom, Belgium). The plants used in these experiments were at the vegetative development stage of four weeks after seedling for *A. hybridus* and *O. basilicum* and six weeks after seedling for *O. gratissimum*.

*Myzus persicae* and *A. fabae* were maintained on a broad bean plant–*V. faba* L. *Aphis craccivora* were collected on an *A. hybridus* crop in Libreville, Gabon (0°27′30.46″ N; 9°25′6.30″ E) and were subsequently reared on *A. hybridus* plants. Aphid species were separately reared in $45 \times 45 \times 45$ cm net cages (BugDorm, MegaView Science, Taichung, Taiwan) under the same laboratory conditions.

### 2.2. Impact of Ocimum on the Dispersion of Unwinged Aphids

Three treatments were tested for each aphid species: (1) *A. hybridus* alone (control), (2) *A. hybridus* + *O. gratissimum* (Ah + Og), and (3) *A. hybridus* + *O. basilicum* (Figure 1). For each treatment, one plant (*A. hybridus*, *O. gratissimum* or *O. basilicum*) was set in the center of a tray. Two batches of four plants were distributed at 8 cm and 12 cm from the central plant (Figure 1). Twenty (wingless) adult aphids were transferred to the central plant (release plant) using a fine brush. The plants located at 8 and 12 cm from the central plant were considered as a potential selected plant by aphids. The trays were

individually placed in the 45 × 45 × 45 cm net cages (BugDorm, MegaView Science, Taichung, Taiwan). Twelve days later, the number of aphids on the central and selected plants at 8 cm ($n = 4$) and 12 cm ($n = 4$) were counted. Six replicates were assessed per treatment and aphid species. The repellent index (RI) was calculated using the following formula adapted from [29]:

$$RI = (P\_sp - P\_cp)/(P\_sp + P\_cp), \tag{1}$$

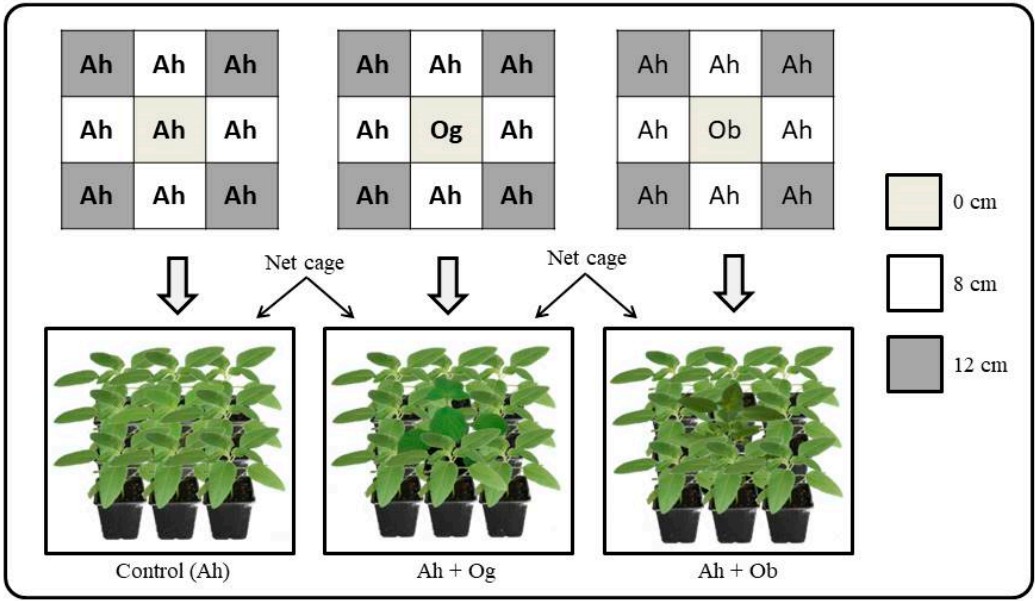

**Figure 1.** Illustration of the different treatments. Ah: *A. hybridus*, Og: *O. gratissimum*, Ob: *O. basilicum*, control (*A. hybridus* alone), 8 cm and 12 cm: distance to the central plant (0 cm).

Psp: percentage of aphids on selected plant (plants at 8 cm and 12 cm), Pcp: percentage of aphids on central plant (*A. hybridus* or *Ocimum*). Repellent index ranging from -1.00 to 1.00. Negative value indicates attractant effect toward the central plant, and positive value indicates a repellent effect to the central plant.

### 2.3. Impact of Ocimum on the Behavior of Winged Aphids in the Dual-Choice Test

A flight tunnel (90 × 45 × 45 cm) was used to evaluate how *Ocimum* plants in the vicinity of *A. hybridus* plants impacted aphid behavior. The tested modalities were: (1) an *A. hybridus* plant associated with an *Ocimum* plant (*O. gratissimum* or *O. basilicum*) versus an *A. hybridus* alone, and (2) an *A. hybridus* plant versus an *A. hybridus* (black control). Ten replicates were assessed for each modality. For each replicate, twenty winged aphids, randomly sampled from the rearing population, were released in the central area of the tunnel. After 24 h, the number of aphids on each *A. hybridus* plant associated with and without the *Ocimum* plant was recorded.

### 2.4. Statistical Analysis

For the dispersal test, the aphid populations recorded on the selected (8 and 12 cm) and central (0 cm) plants were grouped by treatment: control (*A. hybridus*), *A. hybridus* + *O. gratissimum* (Ah + Og) and *A. hybridus* + *O. basilicum* (Ah + Ob). In order to compare the effect of the interactions between the different factors and their main effect, Generalized Linear Models ("glm"), Negative Binomial Generalized Linear Model ("glm.nb") and Zero-Inflated Count Data Regression ("zeroinfl") were compared using the value of their Akaike Information Criterion (AIC), in order to determine the best-fitted model to the data. Following this comparison, the Negative Binomial Generalized Linear

Model (function "glm.nb", "package MASS"; [30]) was used for each level of analysis with aphid species, plant position, plant species and treatment as fixed variables and aphid abundance as the response variable. Two levels of comparison were performed. Firstly, aphid abundance was compared between the selected plants and the central plant for each treatment, and then the number of aphids was compared between the treatments per selected plant. ANOVA test (function "Anova", package "car"; [31]) was performed and the "SNK.test" function ("agricolae package"; [32]) was used for means comparison. For the Repellent Index, the function "aov" was performed on data. A probability level lower than 0.05 was considered statistically significant. All statistical tests were performed using R software version 3.6.3 [33].

Binomial proportion tests (equal distribution hypothesized) were used to compare the number of aphids on the control plant (*A. hybridus* alone) and the associated treatments (*Amaranthus* versus *Ocimum*) during the dual-choice test, according to each modality.

## 3. Results

### 3.1. Effect of Ocimum Species on the Abundance of Unwinged Aphids and Repellent Activity

Analysis of the interactions between the factors studied on the aphids shows that only "Treatments", "Distance", the interaction "Treatments: Distance" and "Aphids: Treatments: Distance" were statistically significant ($p < 0.05$) (Table 1). Synthesis of analysis of the deviance (Anova) for fixed factors – plants, treatments and distance – and their interactions on abundance of each aphid species on amaranth plants is available in the supplementary materials (Table S1).

**Table 1.** Synthesis of analysis of the deviance (Anova) for fixed factors plants, aphids, treatment and distance and their interactions on aphid's abundance on amaranth plants.

| Factors | Df | Chisq | *p*-Value |
|---|---|---|---|
| Plants | 1 | 0.17 | 0.680 |
| Aphids | 2 | 0.99 | 0.608 |
| Treatments | 2 | 12.74 | 0.001 |
| Distance | 2 | 35.63 | <0.001 |
| Aphids: Treatments | 4 | 2.74 | 0.603 |
| Aphis: Distance | 4 | 1.63 | 0.803 |
| Treatments: Distance | 4 | 627.82 | <0.001 |
| Aphids: Treatments: Distance | 8 | 23.43 | 0.002 |

Aphid abundance in each treatment is presented in Figure 2. For all aphid species, the number of individuals decreased from the central plant (0 cm) to the selected plants (8 and 12 cm) in the control (*A. hybridus* plant alone), while the opposite effect was observed in the association treatments (Ah + Og or Ah + Ob). Additionally, no aphid was recorded on an *Ocimum* plant. The means comparison of aphid populations between distances (the central plant and selected plants) for each treatment showed an overall significant effect ($p < 0.05$) (Figure 2). The comparison of aphid abundance between treatments shows that on the central plant (0 cm) the number of aphids was significantly higher in the controls than in the associated treatments (Ah + Og or Ah + Ob). The selected plants closer to the central plant showed no significant difference in aphid numbers compared to the central plant. However, the plant at 12 cm away showed significantly higher numbers of aphids when the central plant was a basil species.

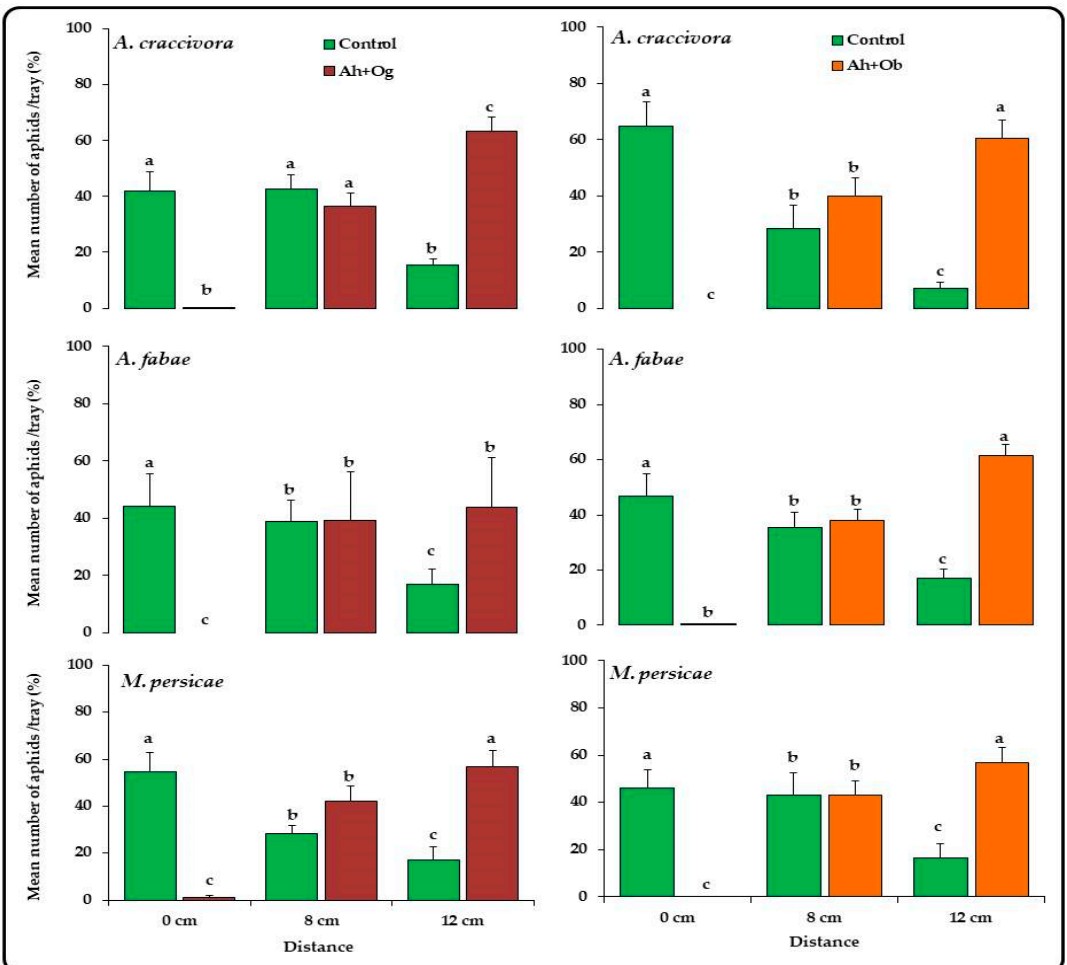

**Figure 2.** Aphid number (Mean ± SE, six replicates) on the central plant (0 cm) and selected plants (8 and 12 cm) according to the tested treatments, twelve days after aphid release. Control: *A. hybridus* alone, Ah: *A. hybridus*, Og: *O. gratissimum*, Ob: *O. basilicum.* For each distance, the bars with the same letter do not differ significantly ($p > 0.05$). Within each treatment (control, Ah + Og and Ah + Ob), the bars with the same letter do not differ significantly ($p > 0.05$).

The repellent indexes of *Ocimum* plants on aphids are presented in Figure 3. In the presence of basil plants (Ah + Og and Ah + Ob), the repellent index is positive, and close to one, irrespective of the distance from the released plant (central plant) and the aphid species. On the other hand, in the absence of the basil plant (control), the repellent index is negative overall, but varied with aphid species. Statistical analysis showed a significant difference between the control and treatments (Ah + Og and Ah + Ob) irrespective of the distance ($p < 0.001$). The synthesis of analysis of the Ocimum repellent index on aphids at 8 and 12 cm is available in the supplementary materials (Table S2).

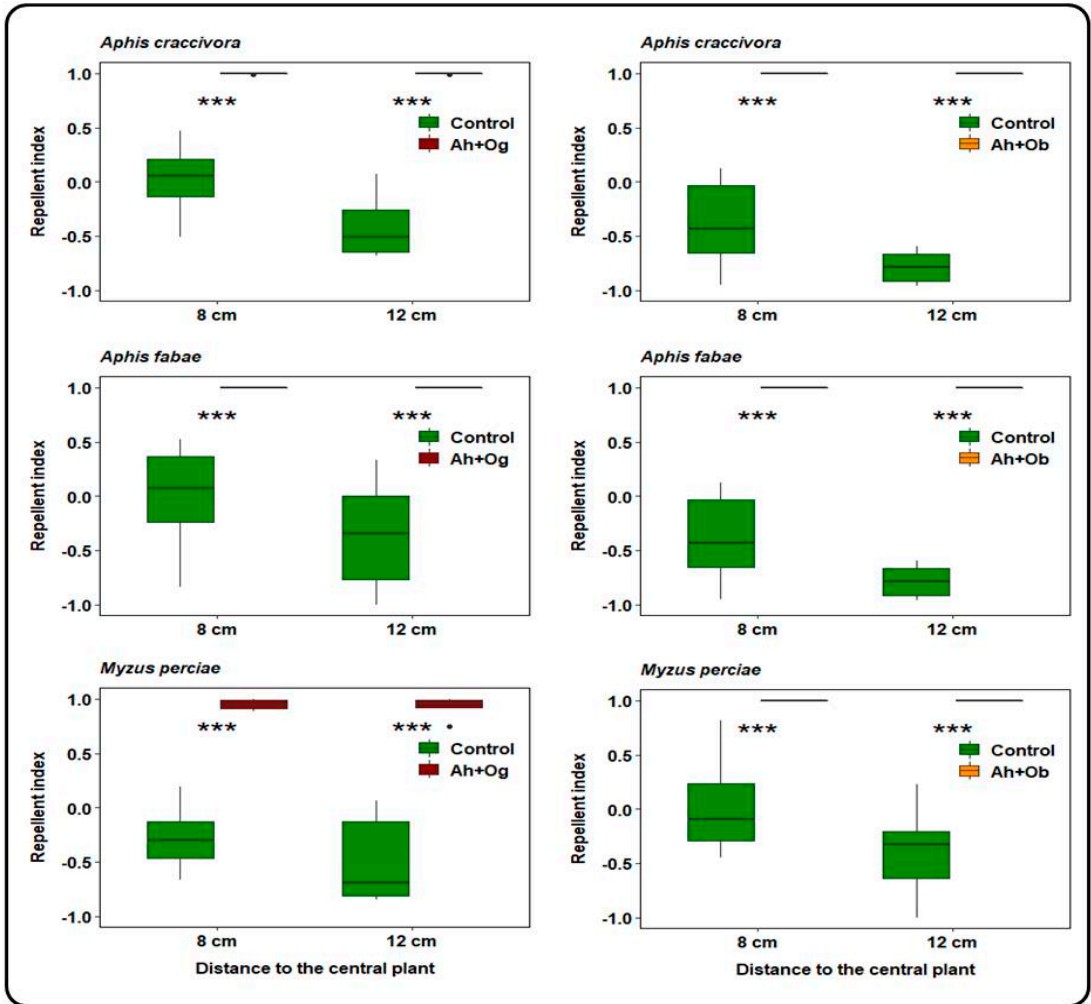

**Figure 3.** Repellent index towards three aphid species according to the selected plants (8 and 12 cm) compared to the central plant (0 cm) in each modality. Control: *A. hybridus* (Ah) alone, Og: *O. gratissimum*, Ob: *O. basilicum*. *** $p < 0.001$.

### 3.2. Repellent Activity of Ocimum on Winged Aphids

Two hundred aphids were tested for each modality. In general, more than 75% of the aphids responded (Table 2). Aphids were evenly distributed among *A. hybridus* when tested alone (Figure 4). In comparison, the number of aphids observed on *A. hybridus* plants associated with *Ocimum* plants (*O. gratissimum* or *O. basilicum*) was significantly less significant ($p < 0.001$) (Figure 4).

**Table 2.** Responding aphids in choice test according to treatment. Control: Ah versus Ah, Ah: *A. hybridus*, Og: *O. gratissimum*, Ob: *O. basilicum*.

| | Treatments | | | | | |
|---|---|---|---|---|---|---|
| | **Control** | | **Ah Versus Ah + Og** | | **Ah Versus Ah + Ob** | |
| **Aphids** | Responding aphids (%) [a] | *p*-value | Responding aphids (%) [a] | *p*-value | Responding aphids (%) [a] | *p*-value |
| *A. craccivora* | 166 (83) | <0.001 | 182 (91) | <0.001 | 167 (84) | <0.001 |
| *A. fabae* | 155 (78) | <0.001 | 158 (79) | <0.001 | 163 (82) | <0.001 |
| *M. persicae* | 167 (84) | <0.001 | 166 (83) | <0.001 | 178 (89) | <0.001 |

[a] Responding insects include living individuals present in one of the two side areas of the tunnel.

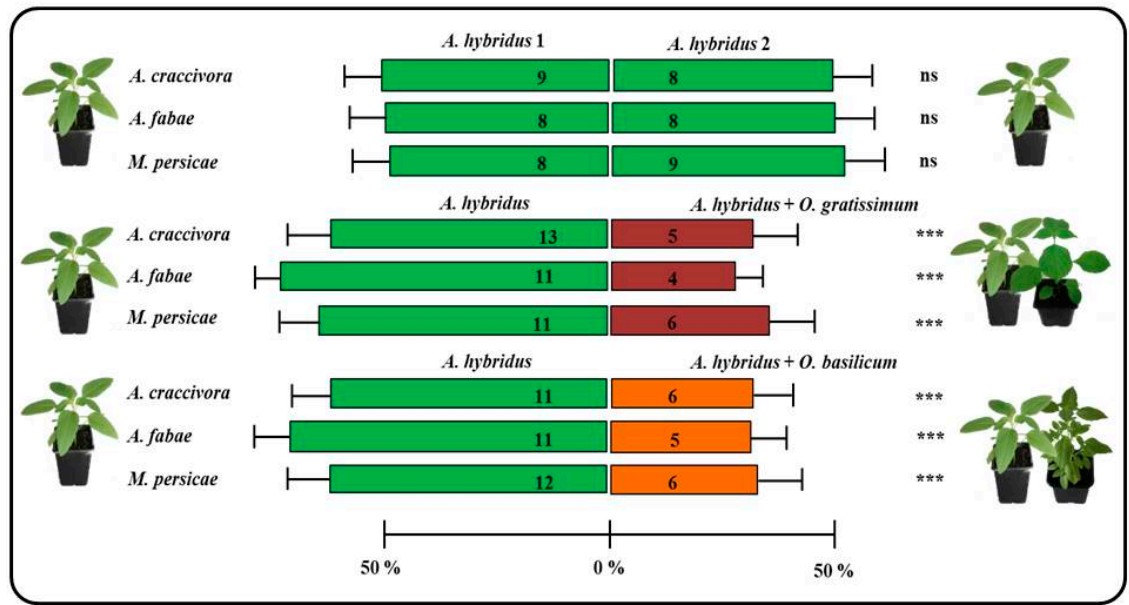

**Figure 4.** Dual-choice of aphids on an *A. hybridus* plant associated with *O. gratissimum* or *O. basilicum* plants or not 24 h after release in the flight tunnel (mean percentage ± SE, 10 replicates). ns: no significant ($p > 0.05$), *** $p < 0.001$.

## 4. Discussion

Pesticidal plants are generally an environmentally friendly solution and are more sustainable in integrated pest management programs than synthetic pesticides. The repellent activity of *O. gratissimum* and *O. basilicum* plants on aphids was determined, as the number of aphids in the control was higher on the central plant. In addition, in the treatments (Ah + Og and Ah + Ob), no aphid was observed on the central plant (*Ocimum* plant) but only on *A. hybridus* plants. This absence of aphids on *Ocimum* plants could firstly indicate an antifeeding or irritant effect of these plants on the aphids, which is defined as a category of repellent effect based on the physiological process of the insect [34]. In this case, a higher number of aphids would probably be observed on *A. hybridus* plants located at 8 cm than at 12 cm from the *Ocimum* plants, because they would only look for a feeding resource. However, overall there were more aphids at 12 cm than 8 cm, with a significant difference in some cases. Additionally, the repellent indexes clearly indicated that this *Ocimum* species had a significant repellent activity on the tested aphids, and this is supported by the choice test results. A similar study reported that the *O. basilicum* plants reduced the infestation level of *Vicia faba* (Fabaceae) by *A. fabae* when these plants were intercropped [19]. The repellent activity of various *Ocimum* species has been reported on other pest families, including Chrysomelidae, Pyralidae, Psyllidae, Pseudococcidae, Tetranychidae, Gelechiidae, Noctuidae, and Plutellidae [9,17,20,21,35]. Other studies have also demonstrated the repellent effect of *Ocimum* extracts or essential oils on many storage pests [12,36] and human disease vectors [13,37].

Host plant location by insects for feeding or reproduction is affected by their ability to perceive VOCs emitted by these plants [38]. However, when host plants are located near to non-host plants, they may be less attractive to insects [39]. Indeed, VOCs from non-host plants can mask or disrupt the chemical environment of the host plants, which would prevent their recognition by pests [34,40]. Thus, it can be assumed that volatiles emitted by *Ocimum* spp., which is considered as a non-host plant, had changed the chemical environment of the amaranth, and then caused the observed repellent effect. According to some studies, the biocidal activity of plant volatiles was generally attributed to their main compounds [41,42]. Therefore, the repellent activity on aphids could be attributed to the major compounds p-cymene, γ-terpinene, α-terpinene, α-thujene E-α-bergamotene, methyl eugenol, E-β-ocimene linalool, previously identified in the volatiles collected on the tested *Ocimum* species [17].

For example, the repellent effect of linalool has been shown on *Cavariella aegopodii* Scopoli, *M. persicae* and *Rhopalosiphum maidis* Fitch [42,43]. The two *Ocimum* species had a similar effect on the three tested aphid species. As they had quite different chemical profiles, according to the relative proportions [17], this could be advantageous in management schemes, as these could be used alternatively to avoid insect habituation.

## 5. Conclusions

Our results provide evidence that Ocimum plants have a repellent effect on aphids. This indicates that these plants have potential as natural pesticides. These plants could then be used as an agroecological alternative for aphid control, in order to avoid the use of synthetic pesticides on amaranth or other vegetable crops in West Africa especially. The ability of *Ocimum* spp. to repel pests and attract natural enemies can make them important companion plants, especially for small-scale farmers. Not only can they be used to control pests, they can also be harvested and commercialized, which may provide a significant economic return for farmers. These benefits can also contribute to the food safety of producers and the consumers. However, additional experiments should be carried out in the field conditions to evaluate the real repellent activity of basil on aphids and how to maximize this effect.

**Supplementary Materials:** The following are available online at http://www.mdpi.com/2073-4395/10/5/736/s1, Table S1: Synthesis of analysis of the deviance (Anova) for fixed factors-plants, treatments and distance-and their interactions on abundance of each aphid species on amaranth plants, Table S2: Synthesis of analysis of the Ocimum repellent index on aphids at 8 and 12 cm.

**Author Contributions:** Conceptualization, B.B.Y.; methodology, B.B.Y.; software, B.B.Y.; validation, G.C.L., F.F.; formal analysis, B.B.Y.; investigation, B.B.Y.; data curation, B.B.Y.; writing—original draft preparation, B.B.Y.; writing—review and editing, A.H.B.-G., G.C.L., F.J.V., G.C.L. and F.F.; supervision, F.F. All authors have read and agreed to the published version of the manuscript.

**Funding:** This research was funded by Erasmus Mundus Program.

**Acknowledgments:** The authors would like to thank Antoine Boullis for his assistance in setting up the experimental system; Félicien Tosso for his assistance in statistical data analysis; Thomas Bawin, Hervé Kombiéni and Sognigbé N'Danikou for reading the manuscript.

**Conflicts of Interest:** The authors declare no conflict of interest.

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
