# Peer review of "Aphid Behavior on Amaranthus hybridus L. (Amaranthaceae) Associated with Ocimum spp. (Lamiaceae) as Repellent Plants"

_agronomy, doi:10.3390/agronomy10050736_

Round 1
Reviewer 1 Report
Reviewers comments have been appropriately addressed.
Reviewer 2 Report
No further comments
Reviewer 3 Report
Many thanks to the authors for thoroughly addressing my comments.
This manuscript is a resubmission of an earlier submission. The following is a list of the peer review reports and author responses from that submission.
Round 1
Reviewer 1 Report
The objective of the research in the manuscript by Barthelemy et al. ws to evaluate possible protective effects of Ocimum plant species against aphids. The work is interesting, but the English is so poor it is difficult to evaluate satisfactorily. I recommend the authors correct the manuscript and resubmit.
Author Response
Dear reviewer,
Please see the attachment the response of your comments and suggestions.
Best regards

Reviewer 2 Report
The paper is fine, but I suggest a few minor changes. I think that the rationale for the work does not do full justice to the results that are presented. Also, I would suggest some changes to the way some of the data is presented, for the sake of clarity. Finally, there are sections in which the language needs to be edited for clarity.
The introduction to the paper puts a lot of emphasis on the practical application of using basil intercropping as an aphid repellent in an agricultural context, however as the authors point out this is a practice that is already used. The experimental design used shows that the presence of basil drives aphids onto the crop species in question- this is not a very compelling demonstration that the intercropping strategy is beneficial- but I think this is a consequence of the experimental design and not a demonstration that the practice is ineffective. It might help give the paper more impact if the authors focus the introduction and conclusions more on the questions that the paper does directly address- a quantitative measure of the relative repellent activity of different basil species- rather than suggesting it is a demonstration of the utility of the agricultural practice.
Figure 2 shows quite clearly the effect sizes of the experiment and is easy to interpret (although the figure is quite small). However, the significance values associated with this figure are displayed in tables. It is quite difficult to reconcile these two methods of presentation. I think that the tables are unnecessary and could be put in supplemental information, and that the significance values would be much more easily interpreted if they could be displayed on figure 2 (as asterisks or other annotations denoting significant effects). The same is true for figure 3.
The language in some places is rather opaque and difficult to understand- I think it could do with some editing for clarity. In particular the paragraph lines 162 to 167 is difficult to understand.
Line 82- “averagely” does not seem to be the right word here. Perhaps “evenly” or “symmetrically” would be better.
Line 91 “Negative value indicates no repellent activity” while the larger the value the greater the repellent effect. I don’t think it’s quite true to say that a negative value means no repellent effect, it does mean a smaller effect for sure.
Line 145-146 “On the first level of the selected plants (8cm), the number of aphids was not significantly different between treatments” I think this needs to be reworded for clarity i think. ie “The selected plants closer to the central plant showed no significant difference in aphid numbers relative to the central plant, however, the further away plants, at 12cm, showed significantly higher numbers of aphids when the central plant was basil”.
Author Response

(The authors gave the same response as above.)

Reviewer 3 Report
It does not harm to mention that the following related study which investigates the potential of biological pest-control concerning e.g. Spodoptera litura:
Lundström, Niklas LP, Hong Zhang, and Åke Brännström. "Pareto-efficient biological pest control enable high efficacy at small costs." Ecological Modelling 364 (2017): 89-97.
Author Response

(The authors gave the same response as above.)

Reviewer 4 Report
Overall, this is a nice paper that is well written – well done! There are a few areas in which the grammar could be improved, I have highlighted these below. Besides this my only other concern is with the number of tables / figures presented in the results section, I feel this could be a little more concise. This is not a major concern, however.
Abstract and Keywords
The abstract provides a good outline of the study and is well written – there is just one small point that requires clarification:
P1, L23: Confused by the statement ‘less important’ on this line – please clarify.
The keywords list the common names for the plant species used in this study, but these common names are then not used in the manuscript text. Please update the manuscript introduction to include the common names. It may also be beneficial to highlight the common names for the aphid species.
Introduction
The introduction is well written and provides good context for study while also clearly outlining the aims. A few minor grammatical edits are required:
P1, L43: Remove comma after ‘pests’.
P1, L44: Add ‘a’ between ‘in’ and ‘semi-field’.
P2, L46: Add ‘a’ between ‘as’ and ‘banker’.
P2, L48-49: Not clear what ‘as dispersal’ means in this context?
P2, L58: Change ‘on orchard ecosystem’ to ‘in an orchard ecosystem’.
Materials and Methods
Again, the materials and methods are well written and demonstrate a good experimental design with appropriate statistical analyses. A few minor grammatical edits are required:
P2, L71: Add ‘conditions’ after ‘greenhouse’.
P2, L73: What growth stage is this – plant age isn’t the most useful indicator.
P2, L80: Add ‘species’ after ‘aphid’.
P2, L82: Add ‘a’ between ‘of’ and ‘tray’.
P2, L82: Remove ‘averagely’.
P2, L84: How were the aphids transferred onto the release plant?
P2, L85: Re-phrase this sentence as it isn’t clear.
P3, L100: Add ‘a’ before ‘flight’.
Results
Although the results are well written, I feel as if there are perhaps too many tables and figures presented for the amount of data. This section could be more concise.
P5, L141: Add ‘an’ between ‘on’ and ‘Ocimum’.
Discussion and Conclusions
Well written and comprehensive, just a few minor grammatical points:
P7, L210: Remove ‘indeed’.
P7, L212: Remove ‘more’.
P8, L240: Add ‘this’ between ‘proportions’ and ‘could’.
Author Response

(The authors gave the same response as above.)
